Manuscript prepared for Atmos. Chem. Phys.
with version 2014/09/16 7.15 Copernicus papers of the LATEX class copernicus.cls.
Date: 30 December 2016

# Uncertainty and variability in atmospheric formation of PFCAs from fluorotelomer precursors

Colin P. Thackray[1] and Noelle E. Selin[1,2]

[1]Department of Earth, Atmospheric and Planetary Sciences, Massachusetts Institute of Technology, 77 Massachusetts Avenue, Cambridge MA 02139 USA
[2]Institute for Data, Systems and Society, Massachusetts Institute of Technology, 77 Massachusetts Avenue, Cambridge MA 02139 USA

*Correspondence to:* Colin P. Thackray (thackray@mit.edu)

**Abstract.** Perfluoroalkyl carboxylic acids (PFCAs) are environmental contaminants that are highly persistent, bio-accumulative, and have been detected along with their atmospheric precursors far from emissions sources. The importance of precursor emissions as an indirect source of PFCAs to the environment is uncertain. Modeling studies have used degradation mechanisms of differing

complexities to estimate the atmospheric production of PFCAs, and these differing mechanisms lead to quantitatively different yields of PFCAs under differing atmospheric conditions. We evaluate PFCA formation with the most complete degradation mechanism to date to our knowledge, using a box model analysis to simulate the atmospheric chemical fate of fluorotelomer precursors to long-chain PFCAs. In particular, we examine the variability in PFCA formation in different chemical

environments, and estimate the uncertainty in PFCA formation due to reaction rate constants.

We calculate long-chain PFCA formation theoretical maximum yields for the degradation of fluorotelomer precursor species at a representative sample of atmospheric conditions from a three dimensional chemical transport model, and estimate uncertainties in such calculations for urban, ocean, and Arctic conditions using polynomial chaos methods. We find that atmospheric conditions farther from

pollution sources have both higher capacities to form long chain PFCAs and higher uncertainties in those capacities.

Our calculations of theoretical maximum yields indicate that under typical Northern Hemisphere conditions, less than 10% of emitted precursor may reach long-chain PFCA end products. This results in a possible upper bound of 2-50 t/yr of long-chain PFCA (depending on quantity of emitted

precursor) produced in the atmosphere via degradation of fluorotelomer products. However, transport to high-yield areas could result in higher yields. While the atmosphere is a potentially growing source

of long-chain PFCAs in the Arctic, oceanic transport and interactions between the atmosphere and ocean may be relatively more important pathways to the Arctic for long-chain PFCAs.

KEYWORDS: PFCA, PFAS, FTOH, perfluoroalkyl, perfluoroalkyl carboxylic acids, per- and polyfluorinated chemicals

## 1 Introduction

Perfluoroalkyl carboxylic acids (PFCAs) are environmental contaminants that are highly persistent, bio-accumulative (Martin et al., 2003a, b; Conder et al., 2008), and have been detected along with their atmospheric precursors far from emissions sources (Young et al., 2007; Shoeib et al., 2006; Stock et al., 2007) in snow (Xie et al., 2015), precipitation (Scott et al., 2006), and biota (Houde et al., 2006). Of particular environmental interest are the long-chain PFCA (lcPFCA, PFCAs of chain length greater than 7) homologues such as PFOA (8-Carbon chain), due to the increased bioaccumulation with chain length (Martin et al., 2003a, b; Conder et al., 2008). PFCAs and their salts are directly emitted to the environment and can be transported long distances via the ocean, having important consequences for remote aquatic biota. While lcPFCAs are not regulated internationally, reducing lcPFCA emissions has been the focus of some national policy actions due to their detrimental health effects (Vierke et al., 2012), and as a result, direct emissions have been decreasing globally. At the same time, emissions of atmospheric precursors of PFCAs are rising (Wang et al., 2014a), leading to an increasing indirect source of PFCAs to the environment. These precursors, including fluorotelomer alcohols (FTOHs), react with atmospheric photochemical species (Ellis et al., 2003) in a multi-stage process to form PFCAs (Young and Mabury, 2010). However, the importance of precursor emissions as an indirect source of lcPFCAs to the environment is uncertain. Estimated yields of PFCAs from precursors can vary based on differences in the formation mechanism assumed, quantitative uncertainty in reaction rate constants, and ambient concentrations of other atmospheric species. Here, we use a box model analysis to quantitatively estimate potential upper-limit atmospheric yields of PFCAs, incorporating uncertainty in the precursor degradation mechanism and variability of atmospheric PFCA formation due to photochemical background conditions.

Previous studies have estimated yields of lcPFCAs from the degradation of FTOHs in the atmosphere (Yarwood et al., 2007; Wallington et al., 2006). However, studies have indicated that other emitted atmospheric precursors exist in the form of other fluorotelomer compounds, perfluoroalkyl sulfonamides (FOSAs), and perfluoroalkyl sulfonamidoethanols (FOSEs) (Young and Mabury, 2010; Wang et al., 2014a, b; Young et al., 2008; Butt et al., 2009). Rate coefficients for the reactions in the PFCA formation mechanism are uncertain, affecting estimated yields. The atmospheric formation of PFCAs depends on reactions of fluorinated intermediates (Waterland and Dobbs, 2007; Chiappero et al., 2006) with commonly studied photochemical species, such as $HO_x$ and

NO$_x$ species, as well as ultraviolet light. These species vary greatly over different environments in the atmosphere, affecting the quantity of lcPFCA produced.

Modeling studies have used degradation mechanisms of differing complexities to estimate the atmospheric production of PFCAs, and these differing mechanisms lead to quantitatively different yields of lcPFCAs under differing atmospheric conditions. Wallington et al. (2006) simulated the atmospheric degradation of 8:2 FTOHs using the IMPACT atmospheric chemistry model, finding that PFOA yields ranged from 1-10% depending on location and time. Yarwood et al. (2007) used a higher resolution atmospheric chemistry model over North America to estimate that degradation yielded approximately 6% PFOA on average, and much less than 1% PFNA. Schenker et al. (2007), using a global-scale multispecies mass-balance model with simplified chemistry, found that precursor transport and degradation could contribute to perfluorocarboxylates observed in the Arctic, and that rate constant uncertainty was an important contributor to uncertainty in their results (Schenker et al., 2007).

In our work, we evaluate PFCA formation with the most complete degradation mechanism to date to our knowledge, including the reactions presented in the studies of Wallington et al. (2006) and Yarwood et al. (2007), and the review of Young and Mabury (2010). Our goal in this work is to examine the variability in PFCA formation in different chemical environments, and estimate the uncertainty in PFCA formation due to reaction rate constants. We use a box model analysis to simulate the atmospheric chemical fate of fluorotelomer aldehyde (FTAL), a common early product in the degradation of many of the different precursor species, including FTOHs. We quantitatively estimate the influence of uncertainty in rate coefficients for calculations of PFCA yields using polynomial chaos methods, which have been used previously in the context of chemical reaction mechanisms (Phenix et al., 1998) and atmospheric chemistry modeling in particular (Cheng and Sandu, 2009; Thackray et al., 2015). We further examine the influence of different atmospheric chemical conditions on upper-limit PFCA formation based on output from a three-dimensional chemical transport model. We conclude by estimating potential upper limits for atmospherically formed PFCAs from emitted precursors, and compare our yield results to observed atmospherically formed PFCAs.

## 2 Methods

We use a box model representation of the gas-phase chemical reactions that lead to atmospheric PFCA formation to calculate yields per unit precursor species. We calculate yields of PFOA (8 Carbons) and PFNA (9 Carbons) from the degradation of 8:2 fluorotelomer precursors. We use prescribed concentrations of photochemical species from data sources described below. To quantify an upper limit of possible atmospheric PFCA formation, we calculate yields of PFOA and PFNA in the absence of non-chemical loss processes, such as sorption to atmospheric particulate matter or removal by wet or dry deposition. Thus, our calculations represent an upper limit of the PFCA for-

mation capacity of the atmosphere at given photochemical conditions. We use the LSODE solver implemented in the "scipy" package of Python to solve the system of differential equations defined by this chemistry.

## 2.1 Mechanism and Box Model

In our box model, we use a precursor degradation mechanism which builds on the work of previous modeling efforts (Wallington et al., 2006; Yarwood et al., 2007) and includes reactions from recent literature (Young and Mabury, 2010). The chemical reactions included are listed in Appendix A, and depicted in Figure 1. The mechanism defines the degradation of fluorotelomer aldehyde, which we use as a generic precursor as it is the first degradation product of emitted volatile fluorotelomer

compounds such as FTOHs and FT-iodides. This FT-aldehyde can be oxidized by OH or photolyzed to form peroxy or acylperoxy radicals. These radicals, in turn, react with NO, $NO_2$, $RO_2$, and $HO_2$ to form stable intermediates. These stable intermediates can again be radicalized by further reaction with OH and ultraviolet light, with more analogous radical reactions leading to either stable PFCAs or shorter chain intermediates. Reaction products which have chain lengths shorter than PFOA are

neglected in our calculations.

We use a box model of the PFCA formation chemistry to calculate yields of PFOA and PFNA from precursor species. The single-box model simulates the chemical reactions discussed above, treating the concentrations of $HO_x$, $NO_x$, Cl, and $RO_2$ as constant and neglecting non-chemical loss processes such as wet and dry deposition. Simulations begin with a unit of precursor species and

110 are carried out until all of the initial precursor has reached one of the reaction end-points (PFNA, PFOA, or shorter-chain PFCAs). The yield of each end species is defined as the fraction of the initial precursor that forms that species.

## 2.2 Variability of PFCA formation

To quantify the variability of PFCA formation capacity due to variations in the atmospheric chemical

background of the Northern Hemisphere, we use photochemical species concentration output from the chemical transport model GEOS-Chem (Bey et al., 2001). We use concentrations of OH, $HO_2$, NO, $NO_2$, and temperature output from a GEOS-Chem version 9.01.02 full chemistry simulation of the years 2006 and 2007 after a one year spin up.

We calculate $RO_2$ concentrations based on concentrations of methane, ethane, and propane from

120 the GEOS-Chem simulation and a pseudo-steady state approximation:

$$[RO_2] \approx \frac{[CH_4][OH]k_{CH_4+OH} + [C_2H_6][OH]k_{C_2H_6+OH} + [C_3H_8][OH]k_{C_3H_8+OH}}{[NO]k_{NO+RO_2} + [HO_2]k_{HO_2+RO_2}} \qquad (1)$$

Available photons for photolysis reactions were calculated based on a scaling by the position of the sun as a function of latitude and time of year and an assumption of clear sky conditions (Russell),

and a peak actinic flux of $1\text{x}10^{15}$ photons cm$^{-2}$ s$^{-1}$ at 0 degrees solar zenith angle (Seinfeld and

Pandis, 2006). We use daily GEOS-Chem output concentrations from winter (January) and summer
(July) of 2007 as a representative sample of the variability of atmospheric conditions in the Northern
Hemisphere.

For the photochemical conditions corresponding with each surface grid box and time of the
GEOS-Chem output, we perform a box model run to calculate yields and formation times of PFOA

and PFNA. This results in 1656 chemical environments for each of the summer and winter condi-
tions.

### 2.3   Uncertainty Propagation

We calculate the parametric uncertainty in yields and formation times for PFCA formation in three
case environments. We use conditions chosen from the above GEOS-Chem output data set repre-

senting three distinct photochemical environments as representative test cases. We have selected one
each of urban, Arctic, and ocean environments for their distinctive PFCA formation behaviors. The
photochemical concentrations of each environment are detailed in Table 1. The urban environment
is located over urban China, and is characterized by high NO$_x$concentrations. The ocean environ-
ment, in contrast, is located over the equatorial Pacific Ocean and is characterized by very low

NO$_x$concentrations. The environment illustrative of Arctic PFCA formation is located over Green-
land, and is much colder and has a moderate level of NO$_x$.

We use polynomial chaos (PC) methods to propagate uncertainty from rate constants to yields cal-
culated by the box model. PC methods create a polynomial expansion representation of the model to
propagate uncertainty in inputs to the outputs at low computational cost while being able to repre-

sent non-linear responses of outputs to model input parameters, as well as interactions between input
parameters (Thackray et al., 2015; Lucas and Prinn, 2005; Cheng and Sandu, 2009). The PC-based
estimator uses orthogonal polynomials to approximate GEOS-Chem model output as a function of
model inputs. The polynomial expansion of the model output to be estimated takes the form

$$\eta(\zeta) = \alpha_0 + \sum_{j=1}^{d}\sum_{k=1}^{M}\alpha_{j,k}H_j(\zeta_k) + \sum_{k=1}^{M-1}\sum_{l=k+1}^{M}\beta_{k,l}H_1(\zeta_k)H_1(\zeta_l) + \ldots + Order(d \geq O > 2) \qquad (2)$$

where the estimator $\eta$ of degree $d$ is a function of the polynomials $H_j$ of order $j$, the $M$ variables
$\zeta_k$ representing model inputs, the expansion coefficients $\alpha_{j,k}$ and $\beta_{k,l}$, and higher order coefficients.
Not shown in the equation are cross terms of degree >2, which include the product of up to $d$ Hermite
polynomials of different variables, analogous to the second order cross terms shown. In this study,
we truncate the polynomial after third order. To obtain the expansion coefficients, one model run at

a unique set of inputs is performed for each term in the equation (Tatang et al., 1997). The set of
inputs for the model runs for each degree's terms are the values corresponding to the roots of the next
degree's polynomials. The outputs of these model runs and the corresponding sets of input values
are used to set up a system of equations to solve for the expansion coefficients (Lucas and Prinn,

2005). We use the polynomial estimator to directly infer properties of the uncertainty distribution
of model output (in this case theoretical maximum fractional yields of PFOA and PFNA) without
relying on Monte Carlo methods, which is accomplished using the analytical forms of the mean and
variance from the polynomial coefficients (Lucas and Prinn, 2005). We also calculate the portion of
the total output variance contributed by each rate constant using the expansion coefficients (Lucas
and Prinn, 2005; Cheng and Sandu, 2009). We carry out a second-order expansion in the 40 uncertain
reaction rate constants to calculate uncertainty distributions of PFOA and PFNA yields and attribute
the importance of each reaction rate constant to the resulting parametric uncertainty.

### 2.4  Environment Categorization

In order to categorize the differences in photochemical environments, we use the DBSCAN clus-
tering algorithm (Ester et al., 1996) to find clusters in summer average OH-HO$_2$-NO concentration
space. These three species were chosen because they are the most common non-fluorinated reac-
tants in the modeled chemistry, and because they led to the delineation of the observed behavior in
yield-time space apparent by visual inspection (see Section 3.4). The DBSCAN algorithm is density-
based, clustering based on the proximity of nearest neighbors in the chosen parameter space. The
algorithm requires a priori values for its two parameters, $\varepsilon$, which roughly describes the size of the
"neighborhood" around a datum, and $N_\varepsilon$, the number of other data that must be within that neigh-
borhood to be considered a cluster. The clustering is relatively insensitive to choice of $N_\varepsilon$ (Ester
et al., 1996), but the number of clusters found in the data set depends on the value of $\varepsilon$ chosen. We
choose an $N_\varepsilon$ value of 10 and the $\varepsilon$ value (0.3) that gives the smallest number of clusters >1 for
simplicity in categorization. This results in two major clusters accounting for >85% of the data, with
the remaining data unclustered.

### 3  Results

We calculate the variability in PFOA and PFNA theoretical maximum yields for summer and winter
Northern Hemisphere conditions, and quantify the parametric uncertainty in these theoretical yields
for three representative test cases. We also investigate the distinct chemical regimes in the formation
of PFNA in different regions of the atmosphere under average summer conditions.

### 3.1  Variability in yields due to photochemical environment

Figure 2 shows histograms of theoretical maximum yields of PFOA and PFNA for each of the pho-
tochemical environments from GEOS-Chem output. Each count in the histogram corresponds to a
calculation of yields carried out at the conditions from a single day and Northern Hemisphere grid-
box (latitude-longitude location) from the GEOS-Chem output. For PFOA during the summer, the
majority of photochemical environments result in yields of between 1% and 10%, with approxi-

mately a quarter of the environments yielding <1% and a third of environments yielding between 10% and 30%. During the winter, the peak of PFOA yields remains between 1-10% but many more environments yield <1% and fewer yield >10% compared to during the summer.

PFNA, on the other hand, sees a peak less than 1% during the summer, but shows a third of its environments between 1% and 10%, with a small fraction of environments leading to yields higher than 80%. During the winter, PFNA formation skews toward very low yields of <0.1%. The long tails of PFNA formation environments are discussed further in Section 3.4. Previous work (Yarwood et al., 2007) quantified yields of PFOA and PFNA over the United States, but the extreme capacities to yield lcPFCAs in remote low-$NO_x$ environments have been previously unquantified.

## 3.2 Uncertainty in yields due to rate constant uncertainty

Figure 3 shows uncertainty in PFOA and PFNA yields due to uncertainty in the rate constants in the degradation mechanism. For both species, yields are negligible under the high-$NO_x$ urban conditions in agreement with previous work focused on North America (Yarwood et al., 2007). Under oceanic conditions far from $NO_x$ sources, the PFOA yield is approximately 20%, with an uncertainty range of approximately 3%, and the PFNA yield is more than 80%, with an uncertainty range of approximately 5%. Under Arctic conditions, PFOA yield uncertainty ranges between 18% and 22%, and PFNA shows a distribution ranging from 17% to 20%. For both species, and especially PFNA, the range of yields due to differing photochemical conditions is much larger than the range of yields due to uncertainty at any given conditions. The greater impact of variability compared to uncertainty means that it is quantitatively viable to model the transport and chemical fate of emissions despite a relatively uncertain set of chemical reactions.

## 3.3 Rate coefficient contributions to yield uncertainty

Fractional contributions of individual reactions' rate coefficient uncertainties to the resulting yield uncertainty for PFOA and PFNA formation are summarized in Table 2. Most reactions in the mechanism contribute to uncertainty similarly for PFOA yield under urban conditions, with reaction 16 having the largest contribution. The rate of this reaction between poly-fluorinated peroxy radicals and $RO_2$ radicals to form a poly-fluorinated alcohol is one of the main factors determining whether the yielded product is PFNA or a shorter chain PFCA (including PFOA), which makes it important for the uncertainties in yields for both of those end products. For ocean conditions, reaction rate constants 15, 16, 36 and 37 dominate the contributions to PFOA yield uncertainty. Arctic conditions show reaction 37's rate constant uncertainty also playing a large role, but reaction 34 also makes a substantial contribution. Reactions 15 and 16 represent a branching in the degradation chemistry where fluorinated peroxy radicals can either branch toward PFNA formation or PFOA and shorter chain PFCAs. Likewise, reactions 34, 36, and 37 are at a branching point where shorter peroxy radicals can either react to form PFOA or even shorter chain PFCAs.

PFNA yield uncertainties are dominated by a different subset of the reaction mechanism for the Arctic environment, and see a contribution from a large number of reaction rates for the urban and ocean cases, led by reactions 16 and 2 (reaction of OH with the initial precursor), respectively. In the Arctic, reaction rate constant 16 uncertainty dominates, with reaction 14 (another peroxy radical reaction) also contributing significantly. In summary, we determine for the first time the dominant sources of uncertainties in theoretical maximum yields of PFOA and PFNA, finding that rate constants of reactions of NO and $RO_2$ with poly- and per-fluorinated peroxy radicals are the leading sources in the degradation chemistry.

## 3.4 Regime behavior in PFNA yields and formation times

Figure 4 shows calculated PFNA yield for each GEOS-Chem grid box and associated time of formation for summer conditions, with DBSCAN algorithm clusters in the OH-$HO_2$-NO space of the sample of summer atmospheric photochemical conditions. Two distinct regimes appear in the plotted space, one in which yield is low across formation times, and one in which longer formation times are associated with higher yields. As Figure 4 shows, the clusters in OH-$HO_2$-NO space correspond to regimes of formation for PFNA, and to spatial regions of the atmosphere. Each of the two clusters respectively compose the majority of each of the two regimes in PFNA yield - time of formation space. Figure 4(b) shows that the same clusters also correspond to Arctic and lower-latitude environments, respectively, indicating a distinct photochemical environment for PFCA formation in the Arctic atmosphere that to our knowledge has not been discussed in previous studies. Within the lower-latitude mode, PFNA yield increases with decreasing NO concentrations, with the lowest yields occurring over land in more polluted areas and the highest yields occurring over the oceans far from $NO_x$ sources.

## 4 Discussion

We find a wide variety of theoretical maximum yields for both PFOA and PFNA across the Northern Hemisphere's photochemical environments. With many regions yielding less than 1% of each due to the presence of large enough quantities of $NO_x$, but PFOA yields of up to 40% and PFNA yields of up to 80% in some areas, the specific photochemical environment has a strong effect on the capacity of the atmosphere to yield lcPFCAs from the degradation of emitted precursors. We find that the parametric uncertainty in these theoretical maximum yields depends on the environment as well, but is at most on the order of a few percent, much smaller than the variability caused by the diversity in photochemical environments.

We find two distinct regimes of PFNA formation capacity in the atmospheric environment, which correspond to photochemical environments found in the Arctic and at lower latitudes, respectively. The former shows relatively constant theoretical maximum yields across all of the conditions within

the Arctic, with a large range of formation times that are independent of the yields. The second regime, on the other hand, shows that at lower latitudes there is a large range of both yields and formation times, and that longer formation times are associated with higher theoretical maximum yields. Within this regime, the higher the concentration of NO, the shorter the formation time and the lower the yield capacity. Figure 5 illustrates this behavior, showing the flux through different reactions in the chemical mechanism over the course of a box model run at the conditions of the three representative environments introduced in Section 2.3. The nodes in the diagram represent intermediate or end-product species in fluorotelomer degradation, while the lines represent the reaction fluxes, with the thickness of the lines proportional to the flux. Figure 5(a) and (b) show that at lower latitudes the amount of NO present strongly drives fluxes towards either short chain PFCAs (Urban, high-NO conditions) or long chain PFCAs (Ocean, low-NO conditions). The reactions of peroxy radicals with NO are too fast in the presence of substantial $NO_x$ to allow branching toward PFNA or PFOA formation.

The highest theoretical maximum yields and longest formation times are associated with conditions over the oceans far from sources and far from common photochemical pollution sources. Emissions of lcPFCA precursors into polluted air masses reduces the potential for those precursors to form lcPFCAs. Put another way, emissions of precursors in otherwise less-polluted regions are conducive to more lcPFCA formation per precursor emitted.

The calculations that we present are of lcPFCA theoretical maximum yields, and are the upper limits of PFOA and PFNA formation for given atmospheric conditions. In the atmosphere, non-chemical loss processes that we neglect in our model limit actual lcPFCA yields compared to their theoretical maxima. In the case of PFNA, as the areas with highest theoretical maximum yields are associated with the longest formation times, they will see larger discrepancies between theoretical and actual yields than areas with lower theoretical maximum yields. Although regions far from $NO_x$ sources have the greatest capacity for PFNA formation, they also are most vulnerable to having concentrations of degradation intermediates reduced by wet deposition and scavenging before the degradation has reached an end product (e.g. over the equatorial oceans).

We calculate the theoretical maximum yields of lcPFCAs from precursor degradation under many atmospheric conditions, but the degradation mechanism is indicative of daytime chemistry. In the Arctic during the summer this is not problematic, but in the winter it neglects the possibility of significant nighttime chemistry involving species such as $N_2O_5$ and $H_2O_2$ that to our knowledge has not been studied. Future research could put theoretical or experimental constraints on the possible importance of these reactions.

With respect to theoretical maximum yields in different seasons, winter conditions lead to lower yields of both PFNA and PFOA, sometimes by orders of magnitude. Young et al. (2007) report a similar seasonal dependence from the Devon Ice Cap, with summer concentrations of PFOA and PFNA being an order of magnitude higher than winter concentrations in the accumulated snow profiles. For

the years 2004 and 2005, the average winter PFNA concentration in those snow measurements is 18 times smaller than the average summer concentration, and for PFOA the winter average is 7 times

smaller. In our calculations, those same ratios over the Canadian Arctic are 18 and 10, respectively. As the long-chain PFCA deposited on the Devon Ice Cap is most likely atmospherically generated (Young et al., 2007; Goss, 2008), this suggests consistency between our calculations of PFNA and PFOA theoretical yields and observational evidence of lcPFCA yielded through formation in the atmosphere.

The importance of the photochemical environment to lcPFCA formation, particularly the importance of the presence of $NO_x$, means that future air pollution reductions or increases could impact atmospheric lcPFCA yields. For instance, large reductions in $NO_x$ emissions would lead to more lcPFCA products. However, given our results, we find that $NO_x$ concentration reductions would have to be on the order of magnitude scale to affect theoretical maximum yields significantly.

We estimate uncertainty ranges in theoretical maximum yields for PFOA and PFNA under the ocean case conditions to be 17-22% and 78-85%, respectively, with most of the uncertainty for PFOA stemming from uncertainty in rate constants at a branching point in the degradation mechanism. In the Arctic case conditions, PFOA maximum yield has a similar value and level of uncertainty as for ocean conditions, while PFNA yields have a much lower value and slightly lower level of

uncertainty. Again, under these conditions, the majority of the uncertainty is due to uncertainty in two peroxy radical reaction rate constants at branching points in the mechanism. Better understanding the quantitative relationship between rate constants at these branching points will have the greatest effect on reducing the parametric uncertainty in theoretical maximum yields.

We quantify the parametric uncertainty in theoretical maximum yields, which depend exclusively

on the rate constants. In the atmosphere, where deposition can play an important role in lcPFCA formation, many other sources of uncertainty for yields will arise, such as rates of deposition, frequency of rainout and washout events, solubility and aqueous chemistry of intermediate species, among others. While the uncertainty due to rate constants is quantifiable based on the chemistry used in our calculations, any missing reactions in the degradation chemistry will be unquantifiable. If our

mechanism is incomplete due to currently unidentified reactions, our estimates of uncertainty would underestimate the full uncertainty of the chemistry. Our estimates of the variability of lcPFCA theoretical maximum yields in the atmosphere are also uncertain due to uncertainty in the photochemical conditions used, which are output from the GEOS-Chem model. The uncertainty in GEOS-Chem calculations of photochemical environment is not quantified here, nor is the uncertainty due the

model grid box size's inherent smoothing of photochemical extremes.

The maximum yields calculated above allow us to estimate potential upper limits on the amount of atmospherically produced long-chain PFCAs given the emitted precursor quantities. The current estimate (Wang et al., 2014a) of volatile 8:2 fluorotelomer compound global releases has an upper bound of 500 t/yr for the year 2010, the only year for which such a detailed estimate is available. Given

the theoretical maximum yields we have calculated, this translates to 50 t/yr of lcPFCA produced
      atmospherically based on median yield values from our calculations. This may be an overestimate,
      however, considering the spatial distribution of theoretical maximum yields. In regions that precur-
      sors are emitted (over continental North America, Europe and Asia), theoretical maximum yields are
      less than 1%. If the precursors and intermediates reside in this type of environment for extended pe-
riods of time, the upper limit of atmospheric lcPFCA production could be 5 t/yr or lower. However,
      larger yields can result when precursors are transported to higher-yield environments. These esti-
      mates of upper limit atmospheric production scale linearly with emissions, so emissions rates lower
      than the upper bound estimates would lead to correspondingly lower atmospheric production max-
      ima. Depending on how long precursor and intermediate species reside in the different atmospheric
regions and the distribution of emissions, yields of lcPFCAs can vary greatly.To illustrate this point,
      Figure 6 shows the time series of the fate of a unit of fluorotelomer precursor released from the
      eastern U.S. and following a trajectory calculated by the HYSPLIT dispersion model, through our
      photochemical environments. Starting in a relatively high $NO_x$ environment, the precursor is quickly
      reacted and short-chain compounds form quickly at the beginning. As the parcel of air is transported
over the Atlantic Ocean and poleward, long chain PFCAs begin to form more quickly. The remain-
      ing intermediates at the end of this period have the potential to form much more PFOA and PFNA
      depending on the future fate of the air parcel. Despite emission into a very low-maximum yield
      environment, the transport is sufficiently fast to allow long-chain PFCA formation.

      We quantify the theoretical maximum yields of formation of lcPFCAs from fluorotelomer pre-
cursor, but there are other precursors that follow different degradation schemes and would therefore
      yield PFCAs in different quantities for the same environment. Precursors such as FOSAs and FOSEs
      are found along with FTOHs in the remote atmosphere (Shoeib et al. 2006) and are also precursors to
      PFOS. Some fluorotelomer precursors such as fluorotelomer olefins follow only a subsection of our
      reaction mechanism because of their structure, and would have higher theoretical maximum yields.
The uncertainty and variability estimates that we present indicate quantitatively that the most
      important piece of information for calculating atmospherically formed PFCAs is their photochem-
      ical environment, and that explicitly accounting for transport in the atmosphere on top of chem-
      istry would give accurate estimates of yielded PFCAs despite uncertainty in the rates of the chem-
      istry involved. This means that the approach of previous studies that use spatially resolved models
(Wallington et al., 2006; Yarwood et al., 2007) is one the most important to our understanding of
      atmospherically generated PFCAs and should be continued in the future. Our results also show,
      however, that accounting only for regional-scale transport as in Yarwood et al. (2007) could miss
      an important fraction of the atmospherically formed long-chain PFCAs, since the capacity for re-
      mote atmospheric conditions to form them is so high. Continued quantitative study of the chemistry
of atmospheric PFCA formation, through updating the chemical mechanism, by accounting for the
      changes in the photochemical environment brought on by synoptic variability, and accounting for

anthropogenic emissions changes relevant to both $HO_x$-$NO_x$ photochemistry and PFCAs themselves has further value over the previous work.

Future calculations with a detailed chemical transport model that also accounts for both deposition processes and transport in the atmosphere would allow for a best estimate of total lcPFCA production in the atmosphere over time. While the U.S. EPA Stewardship Program strives to greatly reduce lcPFCA precursors emitted due to American manufacturers, there remains the possibility of growth of precursor production in Asia in the future, meaning that atmospheric lcPFCA formation could become increasingly important as a source globally and to the Arctic. In the future, if production does shift to shorter chain fluorotelomer products, our findings will apply to correspondingly shorter chain PFCAs formed in the atmosphere, as the chemistry studied is analogous across the homologue series. With the assumption that relative rates at the branching points do not depend on chain length, our calculations can be extended to longer and shorter precursor homologues and correspondingly longer and shorter product homologues. If Y(9) and Y(8) are our calculated maximum yields for PFNA and PFOA, respectively, then the fraction $f_{PFCA}$ of PFCA formation from the "unzipping" step of the mechanism is

$$f_{PFCA} = \frac{Y(8)}{(1 - Y(9))}. \tag{3}$$

Knowing this fraction, yield calculations can be extended to shorter and shorter chain PFCA products using the formula

$$Y(X) = f_{PFCA}(1 - \sum_{i=x+1}^{longer} Y(i)) \tag{4}$$

where the theoretical maximum yield at a given product chain length can be calculated based on the yields of the longer chain products in a given environment.

As an example, Table 3 shows the extension of the Arctic case where the theoretical maximum yields of PFNA and PFOA are 18% and 20%, respectively.

Wallington et al. (2006) estimated 0.4 t/yr of PFOA entering the Arctic due to atmospheric production via 8:2 FTOH degradation; the amount entering the Arctic is less than half of global atmospheric PFOA production. This was calculated assuming 1000 t/yr of FTOH emitted to the atmosphere, which is twice the current upper bound of total fluorotelomer emissions to air. Wania (2007) estimated that the amount of atmospherically generated PFCAs deposited in the Arctic peaked in 2005 at 0.154 t/yr, and that 11-21 t/yr is transported to the Arctic via the ocean. Both of these studies estimate atmospherically generated quantities of lcPFCAs which fall reasonably beneath our calculated theoretical maxima. Our results indicate, however, that the region over the oceans is the leading atmospheric environment for lcPFCA formation, meaning that transport to the Arctic via the ocean can be importantly affected by lcPFCAs formed atmospherically at lower latitudes. A detailed coupled atmosphere-ocean model could give important insights to future studies. We quantify variability in atmospherically formed PFCAs but direct emissions and transport of PFOA and its salts are also en-

vironmentally relevant, as transport to remote regions through the ocean has historically likely been dominated by these direct emissions (Wania, 2007).

## 5    Conclusions

We calculate PFOA and PFNA formation theoretical maximum yields for the degradation of precursor species at a representative sample of atmospheric conditions, and estimate uncertainties in such calculations for urban, ocean, and Arctic conditions. We find that atmospheric conditions farther from pollution sources have both higher capacities to form long chain PFCAs and higher uncertainties in those capacities. The greatest uncertainty reductions through reaction rate determinations can

be achieved by better quantifying rate constants at the branching points of the degradation chemistry. We find that there are distinct regimes of PFNA formation behavior in different photochemical environments, dictated by the quantities of $HO_x$ and $NO_x$ species, but less variability in the formation of PFOA.

While we study the daytime chemistry in detail, future studies should investigate the role of night-

time chemistry in lcPFCA formation. The role of non-chemical removal processes from the atmosphere is also an important part of atmospheric lcPFCA formation, and its environmental connection to yields of formation should be investigated.

Our calculations of theoretical maximum yields indicate that most likely less than 10% of emitted precursor can reach lcPFCA end products in the Northern Hemisphere, even ignoring non-chemical

losses. This results in an upper bound of 2-50 t/yr of lcPFCA (depending on quantity of emitted precursor) produced in the atmosphere via degradation of fluorotelomer products. Only a fraction of that is destined to directly deposit in the Arctic. While the atmosphere is a potentially growing source of lcPFCA in the Arctic, oceanic transport of directly emitted, and to a lesser extent low-latitude atmospherically generated, PFCAs are likely more important pathways to the Arctic for

lcPFCA.

*Acknowledgements.*    This work was supported by the U.S. National Science Foundation Arctic Natural Sciences Program (1203526) and a fellowship from the National Science and Engineering Research Council of Canada (to C.P.T.).

**Appendix A: List of reactions**

| Reaction | Rate constant expression | uncertainty | source |
|---|---|---|---|
| **(1)** C8F17CH2C(O)H + hv350 -> C8F17CH2OO | $1.5 \times 10^{-21}$ [cm$^2$ photon$^{-1}$s$^{-1}$] | $7.5 \times 10^{-22}$ | 1 |
| **(2)** C8F17CH2C(O)H + OH -> C8F17CH2C(O)OO | $2.0 \times 10^{-12}$ [cm$^3$s$^{-1}$] | $0.4 \times 10^{-12}$ | 1 |
| **(3)** C8F17CH2C(O)H + Cl -> C8F17CH2C(O)OO | $1.9 \times 10^{-11}$ [cm$^3$s$^{-1}$] | $0.2 \times 10^{-11}$ | 1 |
| **(4)** C8F17CH2C(O)OO + NO2 -> C8F17CH2C(O)OONO2 | $1.1 \times 10^{-11}$(298./T)[cm$^3$s$^{-1}$] | $0.1 \times 10^{-11}$ | 3 |
| **(5)** C8F17CH2C(O)OONO2 -> C8F17CH2C(O)OO | $2.8 \times 10^{16}$exp(T/-13580)[s$^{-1}$] | $0.2 \times 10^{16}$ | 3 |
| **(6)** C8F17CH2C(O)OO + NO -> C8F17CH2OO | $7 \times 10^{-12}$exp(T/340)[cm$^3$s$^{-1}$] | $0.5 \times 10^{-12}$ | 3 |
| **(7)** C8F17CH2C(O)OO + HO2 -> C8F17CH2OO | $3.1 \times 10^{-13}$exp(T/1040)[cm$^3$s$^{-1}$] | $0.3 \times 10^{-13}$ | 3,1 |
| **(8)** C8F17CH2C(O)OO + HO2 -> C8F17CH2C(O)OH | $1.2 \times 10^{-13}$exp(T/1040)[cm$^3$s$^{-1}$] | $0.1 \times 10^{-13}$ | 3,1 |
| **(9)** C8F17CH2C(O)OO + CH3O2 -> C8F17CH2OO | $1.8 \times 10^{-12}$exp(T/500)[cm$^3$s$^{-1}$] | $3.6 \times 10^{-13}$ | 2 |
| **(10)** C8F17CH2C(O)OO + CH3O2 -> C8F17CH2C(O)OH | $2.0 \times 10^{-13}$exp(T/500)[cm$^3$s$^{-1}$] | $4.0 \times 10^{-14}$ | 2 |
| **(11)** C8F17CH2C(O)OH + OH -> C8F17CH2OO | $2.02 \times 10^{-14}$exp(T/920)[cm$^3$s$^{-1}$] | $0.6 \times 10^{-14}$ | 2 |
| **(12)** C8F17CH2C(O)OH + OH -> C8F17C(O)H | $1.13 \times 10^{-14}$exp(T/920)[cm$^3$s$^{-1}$] | $0.32 \times 10^{-14}$ | 2 |
| **(13)** C8F17CH2OO + HO2 -> C8F17CH2OOH | $4.1 \times 10^{-13}$exp(T/750)[cm$^3$s$^{-1}$] | $0.4 \times 10^{-13}$ | 3 |
| **(14)** C8F17CH2OO + NO -> C8F17CH2O | $2.8 \times 10^{-12}$exp(T/300) [cm$^3$s$^{-1}$] | $0.14 \times 10^{-12}$ | 3 |
| **(15)** C8F17CH2OO + CH3O2 -> C8F17CH2O | $1.9 \times 10^{-14}$exp(T/390)[cm$^3$s$^{-1}$] | $0.26 \times 10^{-14}$ | 2 |
| **(16)** C8F17CH2OO + CH3O2 -> C8F17CH2OH | $7.6 \times 10^{-14}$exp(T/390)[cm$^3$s$^{-1}$] | $1.06 \times 10^{-14}$ | 2 |
| **(17)** C8F17CH2OH + OH -> C8F17C(O)H | $1.02 \times 10^{-13}$exp(T/-350)[cm$^3$s$^{-1}$] | $0.1 \times 10^{-13}$ | 4 |
| **(18)** C8F17CH2OH + Cl -> C8F17C(O)H | $6.5 \times 10^{-13}$exp(T/-350)[cm$^3$s$^{-1}$] | $1.0 \times 10^{-13}$ | 2 |
| **(19)** C8F17CH2OOH + OH -> C8F17CH2OO | $4.0 \times 10^{-12}$exp(T/200)[cm$^3$s$^{-1}$] | $1.0 \times 10^{-12}$ | 2 |
| **(20)** C8F17CH2O -> C8F17OO | $2.5 \times 10^{1}$[s$^{-1}$] | $0.1 \times 10^{1}$ | 4 |
| **(21)** C8F17C(O)H + hv350 -> C8F17OO | $1.6 \times 10^{-21}$[cm$^2$ photon$^{-1}$s$^{-1}$] | $0.12 \times 10^{-21}$ | 1 |
| **(22)** C8F17C(O)H + OH -> C8F17C(O)OO | $6.1 \times 10^{-13}$ [cm$^3$s$^{-1}$] | $0.5 \times 10^{-13}$ | 1 |
| **(23)** C8F17C(O)H + Cl -> C8F17C(O)OO | $2.8 \times 10^{-12}$ [cm$^3$s$^{-1}$] | $0.7 \times 10^{-12}$ | 1 |
| **(24)** C8F17C(O)H + H2O -> C8F17CHOHOH | $1.0 \times 10^{-23}$[cm$^3$s$^{-1}$] | | 1 |
| **(25)** C8F17CHOHOH + OH -> C8F17C(O)OH | $1.22 \times 10^{-13}$ [cm$^3$s$^{-1}$] | $0.26 \times 10^{-13}$ | 1 |
| **(26)** C8F17CHOHOH + Cl -> C8F17C(O)OH | $5.84 \times 10^{-13}$ [cm$^3$s$^{-1}$] | $0.92 \times 10^{-13}$ | 1 |
| **(27)** C8F17C(O)OO + NO2 -> C8F17C(O)OONO2 | $1.1 \times 10^{-11}$(298./T)[cm$^3$s$^{-1}$] | $0.1 \times 10^{-11}$ | 3 |
| **(28)** C8F17C(O)OONO2 -> C8F17C(O)OO | $2.8 \times 10^{16}$exp(T/-13580)[s$^{-1}$] | $0.2 \times 10^{16}$ | 3 |
| **(29)** C8F17C(O)OO + NO -> C8F17OO | $8.1 \times 10^{-12}$exp(T/270)[cm$^3$s$^{-1}$] | $0.6 \times 10^{-12}$ | 3 |
| **(30)** C8F17C(O)OO + HO2 -> C8F17C(O)OH | $3.1 \times 10^{-13}$exp(T/1040)[cm$^3$s$^{-1}$] | $0.4 \times 10^{-13}$ | 3,1 |
| **(31)** C8F17C(O)OO + HO2 -> C8F17OO | $1.2 \times 10^{-13}$exp(T/1040) [cm$^3$s$^{-1}$] | $0.4 \times 10^{-13}$ | 3,1 |
| **(32)** C8F17C(O)OO + CH3O2 -> C8F17OO | $1.8 \times 10^{-12}$exp(T/500)[cm$^3$s$^{-1}$] | $3.6 \times 10^{-13}$ | 2 |
| **(33)** C8F17C(O)OO + CH3O2 -> C8F17C(O)OH | $2.0 \times 10^{-13}$exp(T/500)[cm$^3$s$^{-1}$] | $4. \times 10^{-14}$ | 2 |
| **(34)** C8F17OO + NO -> C8F17O | $2.8 \times 10^{-12}$exp(T/300.)[cm$^3$s$^{-1}$] | $1.4 \times 10^{-13}$ | 3 |
| **(35)** C8F17OO + HO2 -> C8F17O | $4.1 \times 10^{-13}$exp(T/500.)[cm$^3$s$^{-1}$] | $0.4 \times 10^{-13}$ | 4 |
| **(36)** C8F17OO + CH3O2 -> C8F17O | $2.7 \times 10^{-12}$exp(T/-470.) [cm$^3$s$^{-1}$] | $1.9 \times 10^{-13}$ | 3 |
| **(37)** C8F17OO + CH3O2 -> C8F17OH | $1.0 \times 10^{-13}$exp(T/660)[cm$^3$s$^{-1}$] | $0.6 \times 10^{-14}$ | 3 |
| **(38)** C7F15C(O)F + H2O(l) -> C7F15C(O)OH | $3.86 \times 10^{-6}$[cm$^3$s$^{-1}$] | $0.7 \times 10^{-6}$ | 1 |

[1](Young and Mabury, 2010), [2]JPL Evaluation2015 using hydrocarbon analog, [3] (Wallington et al., 2006), [4] (Yarwood et al., 2007)

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

**Table 1.** Case environment conditions. Photochemical concentrations in cm$^{-3}$, temperatures in K

|  | Urban | Ocean | Arctic |
|---|---|---|---|
| NO | $2\times10^{10}$ | $1.7\times10^{7}$ | $1\times10^{8}$ |
| OH | $2\times10^{7}$ | $5.4\times10^{6}$ | $1.6\times10^{7}$ |
| NO$_2$ | $2\times10^{11}$ | $5\times10^{7}$ | $1\times10^{8}$ |
| HO$_2$ | $9\times10^{6}$ | $1\times10^{8}$ | $3.7\times10^{5}$ |
| RO$_2$ | $8\times10^{6}$ | $1.6\times10^{9}$ | $2.2\times10^{8}$ |
| h$\nu$ | $9.4\times10^{14}$ | $9.76\times10^{14}$ | $1\times10^{15}$ |
| Temperature | 299 | 299 | 265 |

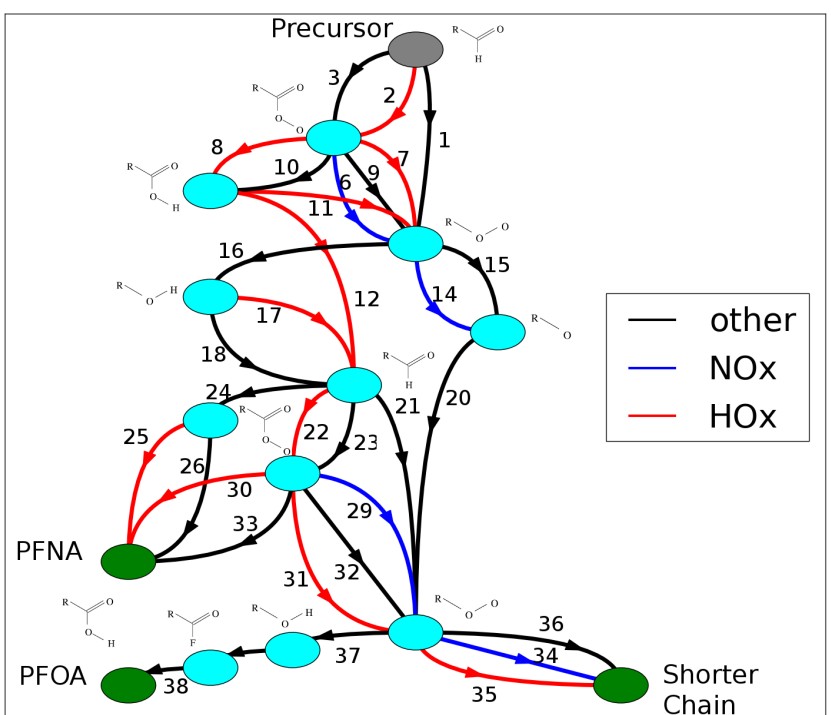

**Figure 1.** Diagram of reaction mechanism used in box model. Each line represents a reaction, with color of the line indicating the photochemical family of the non-fluorinated reactant. Numbers correspond to specific reactions listed in Appendix A.

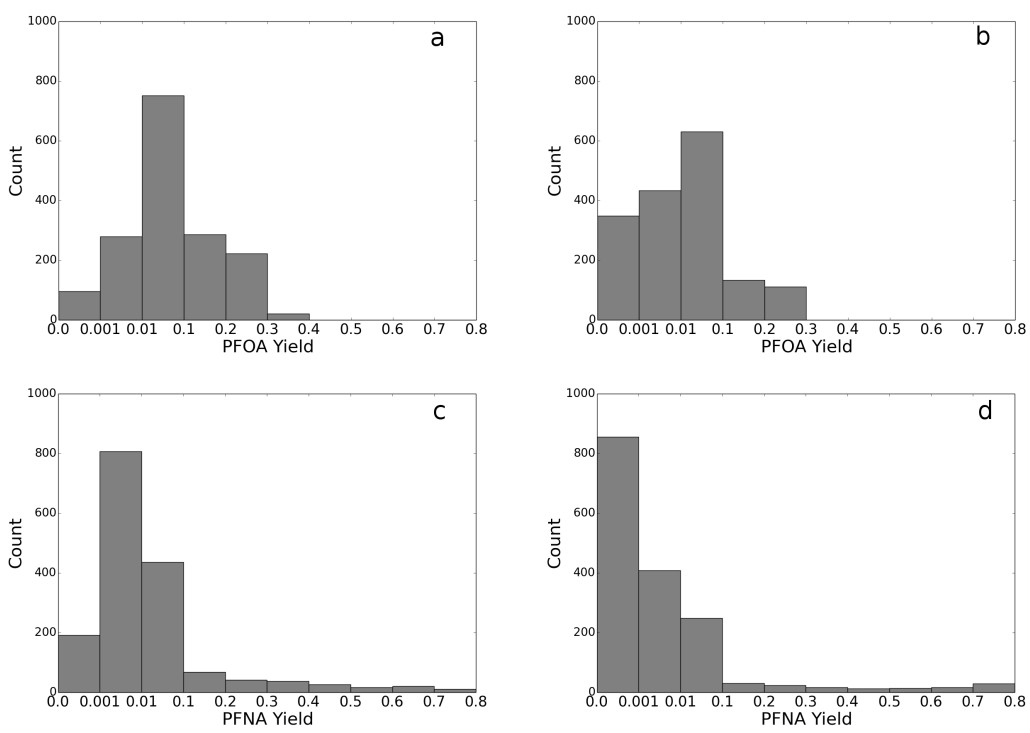

**Figure 2.** Histograms of variability in PFOA (a and b) and PFNA (c and d) theoretical maximum yields for both summer (a and c) and winter (b and d) conditions. Each count corresponds to a GEOS-Chem grid-box's output photochemical environment.

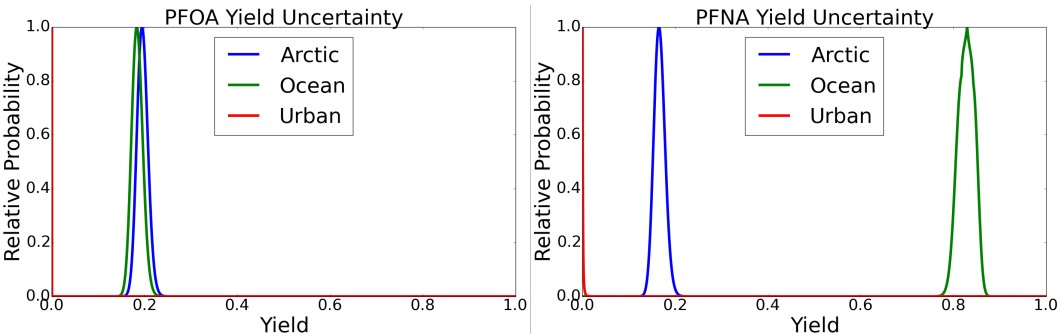

**Figure 3.** Uncertainty distributions of PFOA (left) and PFNA (right) yields for urban, ocean, and Arctic conditions. In both cases, urban yields are much less than 1%

**Table 2.** Fractional yield uncertainty contributions of rate constants (%)

| Rxn # | PFOA urban | PFOA ocean | PFOA Arctic | PFNA urban | PFNA ocean | PNFA Arctic | Reaction |
|---|---|---|---|---|---|---|---|
| 1 | <1 | <1 | <1 | <1 | 2 | <1 | C8F17CH2C(O)H + h$\nu$ -> C8F17CH2OO |
| 2 | <1 | <1 | <1 | 2 | 10 | <1 | C8F17CH2C(O)H + OH -> C8F17CH2C(O)OO |
| 6 | <1 | <1 | <1 | 1 | 2 | <1 | C8F17CH2C(O)OO + NO -> C8F17CH2OO |
| 7 | <1 | <1 | <1 | 1 | 2 | <1 | C8F17CH2C(O)OO + HO2 -> C8F17CH2OO |
| 8 | 7 | <1 | <1 | <1 | 2 | <1 | C8F17CH2C(O)OO + HO2 -> C8F17CH2C(O)OH |
| 10 | 5 | <1 | <1 | 8 | 2 | 4 | C8F17CH2C(O)OO + RO2 -> C8F17CH2C(O)OH |
| 11 | <1 | <1 | <1 | 2 | 2 | 5 | C8F17CH2C(O)OH + OH -> C8F17CH2OO |
| 12 | <1 | <1 | <1 | 2 | 2 | 4 | C8F17CH2C(O)OH + OH -> C8F17C(O)H |
| 14 | <1 | 7 | <1 | <1 | 1 | 27 | C8F17CH2OO + NO -> C8F17CH2O |
| 15 | <1 | 10 | <1 | <1 | 3 | <1 | C8F17CH2OO + RO2 -> C8F17CH2O |
| 16 | 23 | 48 | <1 | 15 | 3 | 57 | C8F17CH2OO + RO2 -> C8F17CH2OH |
| 29 | 6 | <1 | <1 | 9 | 3 | <1 | C8F17C(O)OO + NO -> C8F17OO |
| 30 | <1 | <1 | <1 | 1 | 1 | <1 | C8F17C(O)OO + HO2 -> C8F17C(O)OH |
| 34 | 5 | <1 | 35 | 1 | 3 | <1 | C8F17OO + NO -> C8F17O |
| 35 | 3 | 2 | <1 | 3 | 3 | <1 | C8F17OO + HO2 -> C8F17O |
| 36 | <1 | 13 | <1 | 2 | 3 | <1 | C8F17OO + RO2 -> C8F17O |
| 37 | 8 | 18 | 63 | 2 | 1 | <1 | C8F17OO + RO2 -> C8F17OH |

**Table 3.** Theoretical maximum yield calculations extended to other precursor lengths and product lengths for an environment with PFNA yield of 18% and PFOA yield of 20%.

| Product | 12:2 Precursor | 10:2 Precursor | 8:2 Precursor | 6:2 Precursor |
|---|---|---|---|---|
| PFTrDA | 0.18 | 0.00 | 0.00 | 0.00 |
| PFDoDA | 0.20 | 0.00 | 0.00 | 0.00 |
| PFUnDA | 0.15 | 0.18 | 0.00 | 0.00 |
| PFDA | 0.11 | 0.20 | 0.00 | 0.00 |
| PFNA | 0.09 | 0.15 | 0.18 | 0.00 |
| PFOA | 0.07 | 0.11 | 0.20 | 0.00 |
| PFHeA | 0.05 | 0.09 | 0.15 | 0.18 |
| PFHxA | 0.04 | 0.07 | 0.11 | 0.20 |
| PFPeA | 0.03 | 0.05 | 0.09 | 0.15 |
| PFDA | 0.02 | 0.04 | 0.07 | 0.11 |
| PFPrA | 0.02 | 0.03 | 0.05 | 0.09 |
| TFA | 0.01 | 0.02 | 0.04 | 0.07 |
| Remainder | 0.04 | 0.07 | 0.12 | 0.20 |

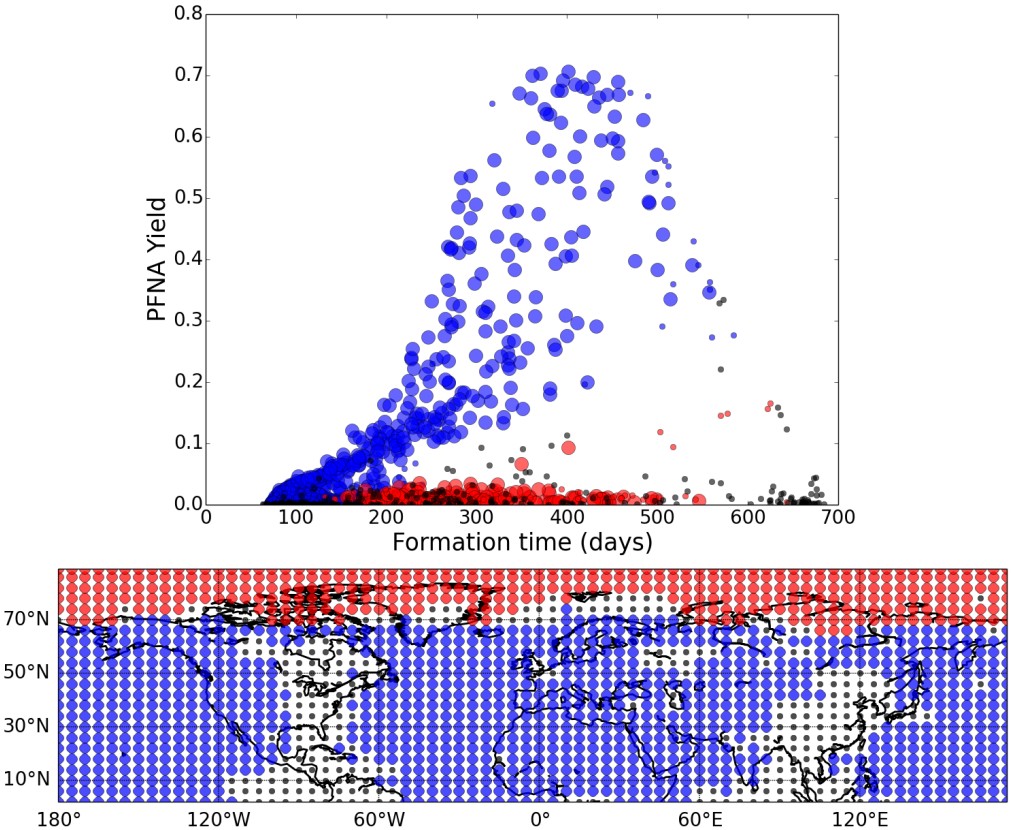

**Figure 4.** (a) Each photochemical environment plotted in yield-formation time space. Color indicates membership of a cluster in OH-HO$_2$-NO space. Black circles indicate unclustered points. (b) Geographic location of clusters. Colors correspond to the same clusters in both figures.

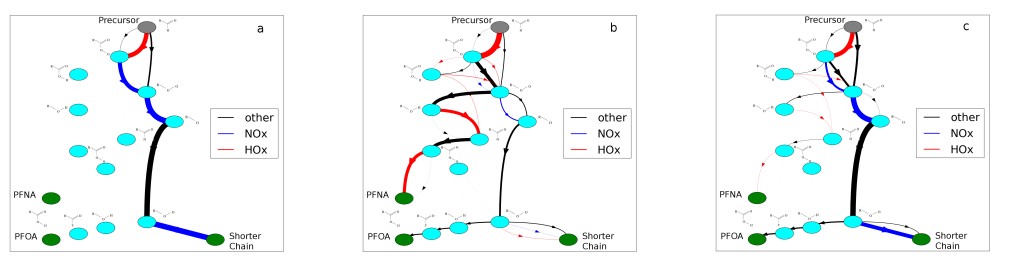

**Figure 5.** Total flux through each reaction for the degradation mechanism for urban (a), ocean (b) and Arctic (c) conditions. Each line represents a reaction, with color of the line indicating the photochemical family of the non-fluorinated reactant, and the thickness of the line is proportional to the total flux through the reaction over the course of a simulation.

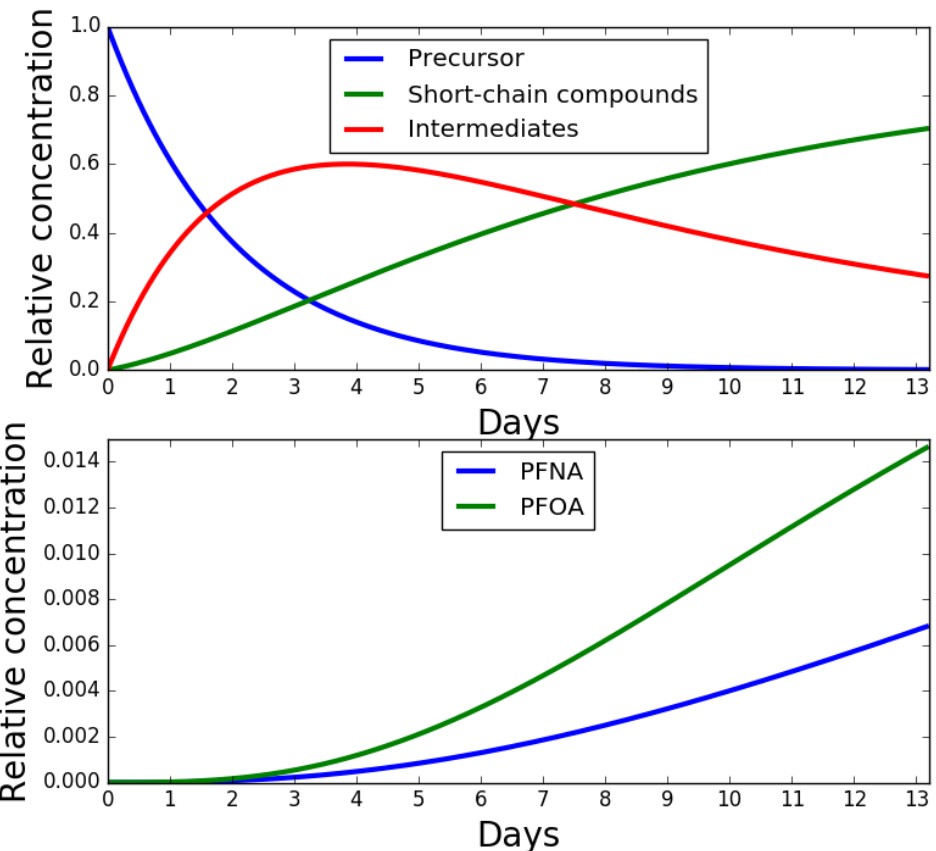

**Figure 6.** Fluorotelomer precursor chemical fate along HYSPLIT trajectory through summertime photochemical environments. After two weeks, yields of PFOA and PFNA are approximately 1.5% and 0.7%, respectively, with more than 20% of the initial precursor still in an intermediate form which will undergo further reactions.