# Peer review of "Uncertainty and variability in atmospheric formation of PFCAs from fluorotelomer precursors"

_Atmospheric Chemistry and Physics, 2016_

## Referee Comment (RC1) · Anonymous Referee #1 · 21 Sep 2016

Summary comments: The title and abstract of the paper were highly promising. In my opinion, however, the paper makes only a small contribution to scientific progress in this field. The authors model the formation of PFOA and PFNA from 8:2 FTOH, which has been done several times before. The authors claim that they have used "...the most complete degradation mechanism to date...". It appears that the photochemical environments and environmental conditions are very well described. However, the authors miss many important precursors of PFOA, including perfluorooctane sulfonamides (FOSAs), perfluorooctane sulfonamidoethanols (FOSEs), 10:2 FTOH etc. and miss to discuss these uncertainties in the discussion. They also miss to discuss the possibilities of direct transport of APFOA (the ammonium salt of PFOA) from polytetrafluoroethylene (PTFE) manufacturing and marine (sea spray) aerosol transport of PFCAs. Even if the authors do not think these processes are important they deserve

discussion. There are atmospheric measurements of FTOH in air from various locations and measurements of PFCAs in precipitation also from various locations. Why were these not discussed or used as a basis for model evaluation? Overall I was very disappointed with the weak discussion; i.e. failure to put the work into proper context. After reading the manuscript I'm not any wiser with regard to the importance of FTOH degradation compared to other sources of PFCAs to the global environment. If there are other similarly negative reviews then the editors may want to consider rejection.

Specific comments: Line 25: perfluoroalkyl carboxylic acids is now the preferred name. Line 28: no year given in Scott et al. Line 29: Define precisely what you mean by "long-chain PFCAs" Line 30: rephrase "increased detrimental effects". I presume the authors mean bioaccumulation potential or toxicity or both? Lines 44-50: Authors miss to discuss other important precursors (see above). Lines 78-84: Are all chemical species assumed to be in the gas phase throughout the reactions? What are the uncertainties generated by this assumption? For example, can reaction intermediates not sorb to aerosols or be rained out? Line 95: "Shorter chain substances" are also generated but it is not specified which or what their yields and formation times are. Why not? This is also highly interesting. Lines 200-215: It is very hard to follow the reaction scheme; an overview figure would have been useful here. Lines 170-230: It would be useful to point exactly what is the novel contribution in the results. What does this study add to previous studies by e.g. Yarwood et al. 2007 nearly a decade ago? They already showed how NOx in populated urban affects PFOA yields. Line 241-242: Explain what you mean by "different conditions within the Arctic"? Lines 300-345. The final discussion excludes many important atmospheric processes including direct atmospheric transport of PFOA following release from manufacturing and marine aerosol transport. The authors also fail to consider atmospheric and deposition measurements of FTOHs and PFCAs. I'm presuming that the model was not able to calculate precipitation scavenging so that comparisons could be made with PFCAs measured in precipitation? Several precursors of PFOA are missing (see above). Note there is evidence that 10:2 FTOH can also form PFOA (see Myers and Mabury (2010) Environmental Toxicology
and Chemistry, Vol. 29, No. 8, pp. 1689–1695). Also it is well known that perfluorooctanesulfonyl fluoride (POSF)-based precursors can form PFCAs and levels of these POSF-based precursors (e.g. FOSEs) have not declined since the 3M phase-out.

---

## Referee Comment (RC2) · Anonymous Referee #2 · 25 Oct 2016

**Summary of manuscript**

The authors investigate, using a chemistry model, the theoretical maximum yield of formation of select long chain perfluorocarboxylic acids (perfluorooctanoic acid, PFOA, and perfluorononanoic acid, PFNA) from precursor species (fluorotelomers). PFOA and PFNA are persistent organic pollutants which bio-accumulate and have detrimental biological effects. The authors use an updated chemical mechanism in a simplified modeling approach (box model vs. spatially resolved atmospheric chemistry model), relative to previous modeling works (Wallington et al., 2006, Yarwood et al., 2007). In the simulations, some loss terms (wet and dry removal) are ignored, hence yields of formation of PFOA and PFNA are theoretical maxima. The authors conduct an interesting analysis of uncertainty propagation which identifies the rate coefficients that have

the largest contribution to the uncertainty in the yields of formation of PFOA and PFNA. Central results of the study are that less than 10 % of emitted fluorotelomer precursors yield PFCAs, and that atmospheric conditions farther from pollution sources (low NOx environments) have both higher capacities to form long chain PFCAs and higher uncertainties in those capacities. With the calculated median theoretical maximum yield from their simulations and a current estimate of global precursor species emissions, the authors estimate the atmospheric production of long chain PFCAs at 50 t/yr.

**Review**

The manuscript has merits, some avoidable oversight errors, and a critical flaw.

The merits include the interesting and useful analysis of uncertainty propagation which identifies the rate coefficients that have the greatest contribution to uncertainty in the yield of formation of the species of interest. Such analysis is useful for laboratory experiments, which can in turn reduce uncertainty of simulations. The analysis of the chemical flux through the reaction mechanism in different environments is instructive and helps increase understanding of the conversion of fluorotelomers to PFCAs. The manuscript is well written, its language is clear and concise.

The critical flaw is the use of a box model. The chemistry simulations are conducted with fixed chemical conditions ("The single-box model simulates the chemical reactions discussed above, treating the concentrations of HOx , NOx, Cl, and RO2 as constant ... until all of the initial precursor has reached one of the reaction end-points (PFNA, PFOA, or shorter-chain PFCAs)."). This neglects changes in chemical conditions that air parcels experience as they are transported.

A box model is appropriate to investigate chemical processes which proceed on time scales that are much shorter than transport time scales. A good example is OH chemistry and certain other chemical processes with time scales that are typically shorter

than a diurnal cycle. In the present work, the authors investigate the conversion of fluorotelomers via fluorotelomer aldehydes (FTAL) to PFOA and PFNA. The chemical scheme in the simulations sets out from FTAL (under the assumption that FTAL forms quickly from the precursor fluorotelomers). FTALs are converted in reactions with OH and Cl (and by photodissociation) to perfluoroacyl peroxy radicals (followed by subsequent transformation towards PFOA and PFNA). The OH and Cl reactions are fairly slow: With the reaction rate coefficients given by the authors and assuming [OH] = 1E6 cm-3, [Cl] = 1E5 cm-3, the corresponding time scales are 5.8 days and 6.1 days, respectively. Transport and mixing are bound to occur on these time scales (the issue is compounded by the very long time of formation of PFOA and PFNA identified the simulations, which exceeds 50 days). The investigation of yields of formation of PFOA and PFNA, a key focus of the present work, makes hence little sense given that air parcels are likely to move away from a location with a specific chemical regime to another on the time scales of the chemistry. The product yields calculated with the chosen approach would reflect reality if air parcels would remain in a given chemical environment longer than the chemical formation of the product, but this seems unlikely.

The issue extends to the analysis of uncertainty propagation from chemistry rate coefficients to product yields. This is the other key focus of the manuscript and one of its interesting parts. In it, the authors determine that it is the reactions of NO and organic peroxy radicals with poly- and perfluorinated peroxy radicals that dominate uncertainty in theoretical maximum yield of PFOA and PFNA. The information is useful for laboratory studies. The identified overall uncertainties are small - theoretical maximum PFOA and PFNA yield ranges (presumably 1-sigma) of 17-22 % and 78-85 % are found. However, given the long formation times from the precursor species to PFOA and PFNA, transport and mixing should be expected to matter - air parcels containing precursor species will experience different conditions on the product formation time scale. The actual product yield may differ from the yield calculated in fixed conditions with a box model. The product yields calculated in the present work hence contain uncertainty introduced by the box model approach. How does this uncertainty compare with the fairly

small uncertainty arising from uncertainty in the rate coefficients? Consider that on the formation time scale of PFOA and PFNA (weeks), an air parcel can experience very different chemical conditions, from highly polluted to oceanic or Arctic. This consideration casts doubt on one of the conclusions of the manuscript, "The greatest uncertainty reductions can be achieved by better quantifying rate constants at the branching points of the degradation chemistry." A more interactive model approach, in which transport and mixing and the associated change in physical and chemical conditions are accounted for could reduce uncertainty to a greater degree than reducing uncertainty in the rate coefficients.

A more interactive model approach (which avoids running a full-fledged atmospheric model) would be to run the chemistry box model along trajectories. Trajectories can be obtained from spatially resolved models using trajectory models such as HYSPLIT or FLEXTRA. It may be possible in this way to extract physical and chemical properties along trajectories from the GEOS-Chem model used by the authors. This approach is more complex than a box-model approach and poses difficulties of its own, but has advantages: Back-trajectories from select deposition regions (such as the Arctic) can be identified and traced back to source regions. The chemistry box model can then be operated with chemical and photochemical input from GEOS-Chem along the trajectories (thereby accounting for change in chemical composition along the trajectories). Thus, one can, in principle, calculate the overall yields on trajectories leading from select emission regions to select deposition regions. The transport issue would be mitigated (although mixing and non-chemical removal would still not be accounted for) and yield attribution to individual sources would become possible.

**Recommendation**

The manuscript contains a critical flaw: A box model with fixed chemical conditions is used to investigate chemical processes that take place on time scales during which which chemical conditions are bound to change due to transport and mixing. I recommend a major revision only if the authors can compellingly demonstrate that the box model approach with fixed conditions is appropriate to investigate formation of PFCAs from from fluorotelomers, despite the formation taking place on time scales during which air parcels are transported and experience different chemical conditions. One way to demonstrate this would be to show that systematically using fixed chemical conditions gives, in reasonable approximation, the results one would obtain if realistic, changing conditions were used. If this is not possible I recommend rejection in favor of a re-submission in which a more appropriate modeling approach, such as the the outlined trajectory approach, is implemented.

**Some detailed comments**

For the benefit of the reader and to facilitate reproducibility, the below comments should be addressed and oversight errors corrected.

Section 2.1

- The numerical solver of the chemistry model should be briefly described.

Section 2.2

- Diurnal cycle: Is it resolved in the simulations, or does the model use perpetual mean conditions, without diurnal cycle variation? Simulations resolving the diurnal cycle would be preferable, being more realistic, but if the latter approach was chosen: how were daily mean photochemistry rates calculated? Was the perpetual mean conditions approach tested by select simulations that do resolve the diurnal cycle, and what were the results? Such a test is inexpensive when a box or a trajectory model is used.

- Actinic flux specification: A value of 1E15 photons cm-3 s-1 at 90 degrees solar zenith angle is given (with reference to Seinfeld and Pandis, 2006). There are several issues which should be addressed: The units of actinic flux are photons cm-2 s-1, not photons cm-3 s-1. Solar zenith angle is measured from zenith: 90 degrees means the sun is at the horizon. This is inconsistent with the specification "peak actinic flux", which, in clear-sky conditions, occurs at noon (corresponding to a solar zenith angle that is typically >= 0 but < 90). Seinfeld and Pandis (2006, Table 4.3) give 340-365 nm mean winter (5E14 cm-2 s-1) and summer (8.9E14 cm-2 s-1) noon actinic flux values at the surface, at 40 degrees north. The actinic flux value used in the simulations is only consistent with the summer value given in Seinfeld and Pandis (2006).

- You write "Available photons for photolysis reactions were calculated as a function of latitude and time of year ..." This should be explained in detail in the manuscript.

Appendix A

- Units of the rate coefficients should be given.

- Fluorotelomer aldehyde photodissociation: I tried to trace the rate coefficient for the reaction 1 (Appendix A), for which the value 1.5+-0.75E-22 (no units) is given, with reference to Young and Mabury (2010). Young and Mabury (2010) give two photodissociation cross sections for FTALs, 13.3E-20 (no error estimate) cm2 (Chiappero et al., 2006) and 5.4+-0.4E-20 cm2 (Solignac et al., 2007), at the maximum of the absorption spectrum. Young and Mabury (2010) do not give the photodissociation rate coefficient. How does the photodissociation rate coefficient 1.5+-0.75E-22 and its error estimate arise?

---

## Author Comment (AC1) · 6 Dec 2016

Throughout this document, reviewer comments are in black and author responses are in blue.

Reviewer 1:

Summary comments: The title and abstract of the paper were highly promising.
We have edited the title and abstract to be more precise about the details of our study.

The title has been changed from "Uncertainty and variability in atmospheric formation of PFCAs" to "Uncertainty and variability in atmospheric formation of PFCAs from fluorotelomer precursors"

We have added the following sentence to the first paragraph of the abstract: "In particular, we examine the variability in PFCA formation in different chemical environments, and estimate the uncertainty in PFCA formation due to reaction rate constants." Line 10 (ca. Line 10 of the original manuscript)

In my opinion, however, the paper makes only a small contribution to scientific progress in this field. The authors model the formation of PFOA and PFNA from 8:2 FTOH, which has been done several times before.
While the reviewer is correct that the formation of PFOA and PFNA from FTOH has been modeled before, we believe our knowledge of these processes can be improved by further modeling. Here, we focus on quantifying the variability in PFOA and PFNA formation in different photochemical environments and estimating the uncertainty in PFOA and PFNA formation due to the rate constants of the reaction mechanism. Neither of these goals has been addressed by previous efforts.

In order to make this clearer, we have changed the abstract to be more precise:
we have added the word "fluorotelomer" to the lines
"We evaluate PFCA formation with the most complete degradation mechanism to date to our knowledge, using a box model analysis to simulate the atmospheric chemical fate of *fluorotelomer* precursors to long-chain PFCAs." Line 6
and
"We calculate long-chain PFCA formation theoretical maximum yields for the degradation of *fluorotelomer* precursor species at a representative sample of atmospheric conditions from a three dimensional chemical transport model…" L9

We have also added the following sentence to the introduction: "Our goal in this work is to examine the variability in PFCA formation in different chemical environments, and estimate the uncertainty in PFCA formation due to reaction rate constants." L66

The authors claim that they have used "...the most complete degradation mechanism to date...".

It appears that the photochemical environments and environmental conditions are very well described. However, the authors miss many important precursors of PFOA, including perfluorooctane sulfon-amides (FOSAs), perfluorooctane sulfonamidoethanols (FOSEs), 10:2 FTOH etc. and miss to discuss these uncertainties in the discussion.

We focus our analysis on 8:2 fluorotelomer precursors, but do not mean to discount other precursors. We have added discussion of how our results apply to fluorotelomer precursors of lengths 10:2 etc. (see below), and have added the following discussion of other precursors, along with further points addressed separately below:

"We quantify the theoretical maximum yields of formation of lcPFCAs from fluorotelomer precursor, but there are other precursors that follow different degradation schemes and would therefore yield PFCAs in different quantities for the same environment. Precursors such as FOSAs and FOSEs are found along with FTOHs in the remote atmosphere (Shoeib et al. 2006) and are also precursors to PFOS. Some fluorotelomer precursors such as fluorotelomer olefins follow only a subsection of our reaction mechanism because of their structure, and would have higher theoretical maximum yields." L326

They also miss to discuss the possibilities of direct transport of APFOA (the ammonium salt of PFOA) from polytetrafluoroethylene (PTFE) manufacturing and marine (sea spray) aerosol transport of PFCAs. Even if the authors do not think these processes are important they deserve discussion.

While the direct emission and transport of PFOA and its salts is important, it is not the focus of this study. To better put this study into context, we have added discussion in response to the reviewer's specific comments below, as well as adding to the introductory paragraph the sentence:

"PFCAs and their salts are directly emitted to the environment and can be transported long distances via the ocean, having important consequences for remote aquatic biota." L31

.

There are atmospheric measurements of FTOH in air from various loca-
tions and measurements of PFCAs in precipitation also from various locations. Why
were these not discussed or used as a basis for model evaluation?

As we discuss in response to comments by Reviewer 2, our calculations of the capacity of different photochemical environments to form PFCAs are not simulations of actual yields of PFCAs at these points, and it would be difficult to infer from our calculations what the concentrations of precursor and PFCAs would be at the location and time of measurements. This kind of comparison is better suited for a model which accounts for emissions, transport, and non-chemical fate of PFCAs, their precursors, and intermediate compounds.

Overall I was very disappointed with the weak discussion; i.e. failure to put the work into proper context.

We hope that after our changes to address the reviewer's specific comments and those of the other reviewer, the discussion has better put this work into context for the reader.

In addition to the changes in response to the reviewer's specific comments below, we have also

added the paragraph:

"The uncertainty and variability estimates that we present indicate quantitatively that the most important piece of information for calculating atmospherically formed PFCAs is their photochemical environment, and that explicitly accounting for transport in the atmosphere on top of chemistry would give accurate estimates of yielded PFCAs despite uncertainty in the rates of the chemistry involved. This means that the approach of previous studies that use spatially resolved models (Wallington et al., 2006; Yarwood et al., 2007) is one the most important to our understanding of atmospherically generated PFCAs and should be continued in the future. Our results also show, however, that accounting only for regional-scale transport as in Yarwood et al. (2007) could miss an important fraction of the atmospherically formed long-chain PFCAs, since the capacity for remote atmospheric conditions to form them is so high. Continued quantitative study of the chemistry of atmospheric PFCA formation, through updating the chemical mechanism, by accounting for the changes in the photochemical environment brought on by synoptic variability, and accounting for anthropogenic emissions changes relevant to both HOx-NOx photochemistry and PFCAs themselves has further value over the previous work." L326

And in response to the other reviewer we have added the following changes relevant to this discussion:
We have added the following discussion (to put our environment categorization into the context of transport through these different environments) and figure after the sentence "Depending on how long precursor and intermediate species reside in the different atmospheric regions and the distribution of emissions, yields of lcPFCAs can vary greatly.":
"To illustrate this point, figure 5 shows the time series of the fate of a unit of fluorotelomer precursor released from the eastern U.S. and following a trajectory calculated by the HYSPLIT dispersion model, through our photochemical environments. Starting in a relatively high NOx environment, the precursor is quickly reacted and short-chain compounds form quickly at the beginning. As the parcel of air is transported over the Atlantic Ocean and poleward, long chain PFCAs begin to form more quickly. The remaining intermediates at the end of this period have the potential to form much more PFOA and PFNA depending on the future fate of the air parcel. Despite emission into a very low-maximum yield environment, the transport is sufficiently fast to allow long-chain PFCA formation."   L326

[Figure]

Figure 5. Fluorotelomer precursor chemical fate along HYSPLIT trajectory through summertime photochemical environments. After two weeks, yields of PFOA and PFNA are approximately 1.5% and 0.7%, respectively, with more than 20% of the initial precursor still in an intermediate form which will undergo further reactions.

After reading the manuscript I'm not any wiser with regard to the importance of FTOH degradation compared to other sources of PFCAs to the global environment. If there are other similarly negative reviews then the editors may want to consider rejection.
We acknowledge the reviewer's point – however, the goal of this paper is not to quantify the importance of FTOH degradation compared to other PFCA sources to the global environment. We hope that our revised abstract and introduction help the reader to better understand the paper's goals.

Specifically, we highlight with this paper the importance of environment for the atmospheric formation of PFCAs and to estimate the uncertainty involved in the chemical mechanism, which has not been done in prior works to our knowledge. While we do not make statements about the importance of fluorotelomer precursors vs. other PFCA precursors, we hope that readers of this study would be wiser with regard to the degradation of fluorotelomer compounds in the atmosphere. To highlight the importance of uncertainty and variability themselves, we have added to the end of the first Discussion paragraph:

"The greater impact of variability compared to uncertainty means that it is quantitatively viable to model the transport and chemical fate of emissions despite a relatively uncertain set of chemical reactions." L196

Specific comments:
Line 25: perfluoroalkyl carboxylic acids is now the preferred name.
We have changed all instances to perfluoroalkyl carboxylic acids

Line 28: no year given in Scott et al.
We have added 2006 to the reference.

Line 29: Define precisely what you mean by "long-
chain PFCAs"
We have added the parenthetical "(PFCAs of chain length greater than 7)"

Line 30: rephrase "increased detrimental effects". I presume the authors
mean bioaccumulation potential or toxicity or both?
We have changed "increase of detrimental effects" to read "increased bioaccumulation"

Lines 44-50: Authors miss to discuss other important precursors (see above).
We have changed the sentence "However, studies have indicated that
other emitted atmospheric precursors exist in the form of other fluorotelomer compounds." to
read "However, studies have indicated that other emitted atmospheric precursors exist in the
form of other fluorotelomer compounds, perfluoroalkyl sulfonamides (FOSAs), and perfluoroalkyl
sulfonamidoethanols (FOSEs)." L45
We have also added discussion of those species that are not the focus of this work, below.

Lines 78-84: Are all chemical species assumed to be in the gas phase throughout the
reactions?
What are the uncertainties generated by this assumption? For example, can reaction
intermediates not sorb to aerosols or be rained out?
Yes, in our model the chemical species are in the gas phase. Removal through non-reaction
processes such as sorption and rainout would dampen the yields compared to our calculations
(if precursor or intermediate) or not affect them (if PFCA end products). For this reason, we call
the result of our calculations "maximum theoretical yields". To clarify our model, we have made
the following changes:

We have added "gas-phase" to the sentence "We use a box model representation of the *gas-phase* chemical reactions that lead to atmospheric PFCA formation to calculate yields per unit precursor species." L78
and have extended the sentence
"To quantify an upper limit of possible atmospheric PFCA formation, we calculate yields of PFOA and PFNA in the absence of non-chemical loss processes." to read
"To quantify an upper limit of possible atmospheric PFCA formation, we calculate yields of PFOA and PFNA in the absence of non-chemical loss processes, such as sorption to atmospheric particulate matter or removal by wet or dry deposition.".L81

Line 95: "Shorter chain substances" are also generated but it is not specified which or what their yields and formation times are. Why not? This is also highly interesting.

Given the assumption of no non-chemical removal processes and making the assumption that the relative rates at branching points are independent of chain length, our calculations can be extended analytically to apply for longer or shorter chain precursors and products. We have therefore added the following text and table to the discussion section:
"In the future, if production does shift to shorter chain fluorotelomer products, our findings will apply to correspondingly shorter chain PFCAs formed in the atmosphere, as the chemistry studied is analogous across the homologue series. With the assumption that relative rates at the branching points do not depend on chain length, our calculations can be extended to longer and shorter precursor homologues and correspondingly longer and shorter product homologues. If Y(9) and Y(8) are our calculated maximum yields for PFNA and PFOA, respectively, then the fraction f_PFCA of PFCA formation from the "unzipping" step of the mechanism is

$$f\_PFCA = Y(8)/(1 - Y(9)) \ .$$

Knowing this fraction, yield calculations can be extended to shorter and shorter chain PFCA products using the formula

$$Y(X) = f\_PFCA \left( 1 - \sum_{i=x+1}^{longer} Y(i) \right)$$

where the theoretical maximum yield at a given product chain length can be calculated based on the yields of the longer chain products in a given environment.

As an example, the table below shows the extension of the Arctic case where the theoretical maximum yields of PFNA and PFOA are 18% and 20%, respectively." L335

Table:

| Product | 12:2 precursor | 10:2 precursor | 8:2 precursor | 6:2 precursor |
|---------|---------------|----------------|---------------|---------------|
| PFTrDA | 0.18 | 0.00 | 0.00 | 0.00 |
| PFDoDA | 0.20 | 0.00 | 0.00 | 0.00 |

| | | | | |
|---|---|---|---|---|
| PFUnDA | 0.15 | 0.18 | 0.00 | 0.00 |
| PFDA | 0.11 | 0.20 | 0.00 | 0.00 |
| PFNA | 0.09 | 0.15 | 0.18 | 0.00 |
| PFOA | 0.07 | 0.11 | 0.20 | 0.00 |
| PFHeA | 0.05 | 0.09 | 0.15 | 0.18 |
| PFHxA | 0.04 | 0.07 | 0.11 | 0.20 |
| PFPeA | 0.03 | 0.05 | 0.09 | 0.15 |
| PFBA | 0.02 | 0.04 | 0.07 | 0.11 |
| PFPrA | 0.02 | 0.03 | 0.05 | 0.09 |
| TFA | 0.01 | 0.02 | 0.04 | 0.07 |
| Remainder | 0.04 | 0.07 | 0.12 | 0.20 |

Lines 200-215: It is very hard to follow the reaction scheme;
an overview figure would have been useful here.
We have added a diagram of the reaction scheme, and corresponding reaction numbers to the
Appendix.

Lines 170-230: It would be useful
to point exactly what is the novel contribution in the results. What does this study add
to previous studies by e.g. Yarwood et al. 2007 nearly a decade ago? They already
showed how NOx in populated urban affects PFOA yields.
To put our contribution into clearer context, we have added/changed the following:
"*Previous work (Yarwood et al., 2007) quantified yields of PFOA and PFNA over the United
States, but the extreme capacities to yield lcPFCAs in remote low-NOx environments have been
previously unquantified.* " Line 187
"For both species, yields are negligible under the high-NOx urban conditions*, in agreement with
previous work focused on North America (Yarwood et al. 2007).*"  Line 190
"In summary, *we determine for the first time the dominant sources of uncertainties in theoretical
maximum yields of PFOA and PFNA, finding that* rate constants of reactions of NO and RO2
with poly- and per-fluorinated peroxy radicals are the *leading sources in the degradation
chemistry.*" Line 217
"Figure 3(b) shows that the same clusters also correspond to Arctic and lower-latitude
environments, respectively, *indicating a distinct photochemical environment for PFCA formation
in the Arctic atmosphere that to our knowledge has not been discussed in previous studies.*"
Line 226

Line 241-242: Explain what
you mean by "different conditions within the Arctic"?
To clarify, we have changed the sentence "The former shows relatively constant theoretical
maximum yields across different conditions within the Arctic, with a large range of formation
times that are independent of the yields." to read
"The former shows relatively constant theoretical maximum yields across *all of the conditions
within* the Arctic, with a large range of formation times that are independent of the yields" L241

Lines 300-345. The final discussion excludes many important atmospheric processes including
direct atmospheric
transport of PFOA following release from manufacturing and marine aerosol transport.
The authors also fail to consider atmospheric and deposition measurements of FTOHs
and PFCAs. I'm presuming that the model was not able to calculate precipitation scav-
enging so that comparisons could be made with PFCAs measured in precipitation?
To the end of the discussion section we have added the following:
"We quantify variability in atmospherically formed PFCAs but direct emissions and transport of
PFOA and its salts are also environmentally relevant, as transport to remote regions through the
ocean has historically likely been dominated by these direct emissions (Wania 2007)." L345
And to the end of the conclusion sentence we have clarified the final sentence to read:
"While the atmosphere is a potentially growing source of lcPFCA in the Arctic, oceanic transport
of directly emitted, and to a lesser extent low-latitude atmospherically generated, PFCAs are
likely more important pathways to the Arctic for lcPFCA." L365

Several precursors of PFOA are missing (see above). Note there is evidence that 10:2
FTOH can also form PFOA (see Myers and Mabury (2010) Environmental Toxicology
and Chemistry, Vol. 29, No. 8, pp. 1689-1695).  Also it is well known that perfluorooc-
tanesulfonyl fluoride (POSF)-based precursors can form PFCAs and levels of these
POSF-based precursors (e.g. FOSEs) have not declined since the 3M phase-out.

We believe that we have addressed these concerns in the responses above.

---

## Author Comment (AC2) · 6 Dec 2016

Throughout this document, reviewer comments are in black and author responses are in blue.

Reviewer 2:

Summary of manuscript
The authors investigate, using a chemistry model, the theoretical maximum yield of formation of select long chain perfluorocarboxylic acids (perfluorooctanoic acid, PFOA, and perfluorononanoic acid, PFNA) from precursor species (fluorotelomers). PFOA and PFNA are persistent organic pollutants which bio-accumulate and have detrimental biological effects. The authors use an updated chemical mechanism in a simplified modeling approach (box model vs. spatially resolved atmospheric chemistry model), relative to previous modeling works (Wallington et al., 2006, Yarwood et al., 2007). In the simulations, some loss terms (wet and dry removal) are ignored, hence yields of formation of PFOA and PFNA are theoretical maxima. The authors conduct an interesting analysis of uncertainty propagation which identifies the rate coefficients that have the largest contribution to the uncertainty in the yields of formation of PFOA and PFNA. Central results of the study are that less than 10 % of emitted fluorotelomer precursors yield PFCAs, and that atmospheric conditions farther from pollution sources (low NOx environments) have both higher capacities to form long chain PFCAs and higher uncertainties in those capacities. With the calculated median theoretical maximum yield from their simulations and a current estimate of global precursor species emissions, the authors estimate the atmospheric production of long chain PFCAs at 50 t/yr.
We thank the reviewer for their thorough reading of our manuscript and their comments below.

The manuscript has merits, some avoidable oversight errors, and a critical flaw.
The merits include the interesting and useful analysis of uncertainty propagation which identifies the rate coefficients that have the greatest contribution to uncertainty in the yield of formation of the species of interest. Such analysis is useful for laboratory experiments, which can in turn reduce uncertainty of simulations. The analysis of the chemical flux through the reaction mechanism in different environments is instructive and helps increase understanding of the conversion of fluorotelomers to PFCAs. The manuscript is well written, its language is clear and concise.
The critical flaw is the use of a box model. The chemistry simulations are conducted with fixed chemical conditions ("The single-box model simulates the chemical reactions discussed above, treating the concentrations of HOx , NOx, Cl, and RO2 as constant ... until all of the initial precursor has reached one of the reaction end-points (PFNA, PFOA, or shorter-chain PFCAs)."). This neglects changes in chemical conditions that air parcels experience as they are transported.
A box model is appropriate to investigate chemical processes which proceed on time scales that are much shorter than transport time scales. A good example is OH chemistry and certain other chemical processes with time scales that are typically shorter than a diurnal cycle. In the present work, the authors investigate the conversion of fluorotelomers via fluorotelomer aldehydes (FTAL) to PFOA and PFNA. The chemical

scheme in the simulations sets out from FTAL (under the assumption that FTAL forms quickly from the precursor fluorotelomers). FTALs are converted in reactions with OH and Cl (and by photodissociation) to perfluoroacyl peroxy radicals (followed by subsequent transformation towards PFOA and PFNA). The OH and Cl reactions are fairly slow: With the reaction rate coefficients given by the authors and assuming [OH] = 1E6 cm-3, [Cl] = 1E5 cm-3, the corresponding time scales are 5.8 days and 6.1 days, respectively. Transport and mixing are bound to occur on these time scales (the issue is compounded by the very long time of formation of PFOA and PFNA identified the simulations, which exceeds 50 days). The investigation of yields of formation of PFOA and PFNA, a key focus of the present work, makes hence little sense given that air parcels are likely to move away from a location with a specific chemical regime to another on the time scales of the chemistry.

We agree that the rate of formation of PFCAs will be much longer than the residence times of given air masses in any one particular environment. We do not wish to convince the reader that the yields we calculate are useful because we believe that the species will sit at these conditions until the precursors become end products, but rather we highlight in a quantitative way the differences in conditions for PFCA formation that the species involved can experience in the atmosphere. Theoretical maximum yields here are a way to quantify the PFCA production capacity of that atmospheric environment, and not intended to match the fate of a single air parcel. We agree that the actual yield per precursor will be determined by mixing and transport likely across many of these environments due to the timescales mentioned, but believe that there is value in quantifying the differences between such environments. In particular, we have made changes to the abstract, introduction, and discussion (in response to Reviewer 1) to better communicate the goals of our analysis:

We have added the following sentence to the first paragraph of the abstract: "In particular, we examine the variability in PFCA formation in different chemical environments, and estimate the uncertainty in PFCA formation due to reaction rate constants." Line 10 (ca. Line 10 of the original manuscript)

We have added the following sentence to the introduction: "Our goal in this work is to examine the variability in PFCA formation in different chemical environments, and estimate the uncertainty in PFCA formation due to reaction rate constants." L66

... we have added to the end of the first Discussion paragraph:
"The greater impact of variability compared to uncertainty means that it is quantitatively viable to model the transport and chemical fate of emissions despite a relatively uncertain set of chemical reactions." L196

The product yields calculated with the chosen approach would reflect reality if air parcels would remain in a given chemical environment longer than the chemical formation of the product, but this seems unlikely. We agree that the theoretical maxima for yields that we calculate do not reflect the reality for any given parcel of air, for the reasons that the reviewer describes. The

theoretical maximum yield here is a quantification of a given set of conditions' capacity for forming PFCAs.  We write "Depending on how long precursor and intermediate species reside in the different atmospheric regions and the distribution of emissions, yields of lcPFCAs can vary greatly."

The issue extends to the analysis of uncertainty propagation from chemistry rate coefficients to product yields. This is the other key focus of the manuscript and one of its interesting parts. In it, the authors determine that it is the reactions of NO and organic peroxy radicals with poly- and perfluorinated peroxy radicals that dominate uncertainty in theoretical maximum yield of PFOA and PFNA. The information is useful for laboratory studies. The identified overall uncertainties are small - theoretical maximum PFOA and PFNA yield ranges (presumably 1-sigma) of 17-22 % and 78-85 % are found. However, given the long formation times from the precursor species to PFOA and PFNA, transport and mixing should be expected to matter - air parcels containing precursor species will experience different conditions on the product formation time scale.
The actual product yield may differ from the yield calculated in fixed conditions with a box model. The product yields calculated in the present work hence contain uncertainty introduced by the box model approach.
We agree that if used predictively as actual yields, our calculated theoretical maximum yields will overestimate and hence contain uncertainty that is greater than that from the rate constants. (See above)

 How does this uncertainty compare with the fairly small uncertainty arising from uncertainty in the rate coefficients? We agree that this is an important point, and as discussed above have added the sentence "The greater impact of variability compared to uncertainty means that it is quantitatively viable to model the transport and chemical fate of emissions despite a relatively uncertain set of chemical reactions." L196

Consider that on the formation time scale of PFOA and PFNA (weeks), an air parcel can experience very
different chemical conditions, from highly polluted to oceanic or Arctic. This consideration casts doubt on one of the conclusions of the manuscript, "The greatest uncertainty reductions can be achieved by better quantifying rate constants at the branching points of the degradation chemistry."
To be more precise and reflect our meaning here, we have changed this sentence to read "The greatest uncertainty reductions *through reaction rate determinations* can be achieved by better quantifying rate constants at the branching points of the degradation chemistry." L351

A more interactive model approach, in which transport
and mixing and the associated change in physical and chemical conditions are accounted for could reduce uncertainty to a greater degree than reducing uncertainty in the rate coefficients.
We agree that a model approach that accounts for mixing and transport and changes in physical

and chemical conditions is important for simulating or predicting the specific fate of given real-world emissions, but do not consider this a reduction of uncertainty, per se, in our context. To make this more clear, we have changed the term "uncertainty" to "parametric uncertainty" in places that it was ambiguous.

A more interactive model approach (which avoids running a full-fledged atmospheric model) would be to run the chemistry box model along trajectories. Trajectories can be obtained from spatially resolved models using trajectory models such as HYSPLIT or FLEXTRA. It may be possible in this way to extract physical and chemical properties along trajectories from the GEOS-Chem model used by the authors.

We think that this would be an excellent way to illustrate the interactions of the different photochemical environments over the lifetime of an air parcel containing precursor species, and have added an example of this to our discussion.

We have added the following discussion and figure after the sentence "Depending on how long precursor and intermediate species reside in the different atmospheric regions and the distribution of emissions, yields of lcPFCAs can vary greatly.":

"To illustrate this point, figure 5 shows the time series of the fate of a unit of fluorotelomer precursor released from the eastern U.S. and following a trajectory calculated by the HYSPLIT dispersion model, through our photochemical environments. Starting in a relatively high NOx environment, the precursor is quickly reacted and short-chain compounds form quickly at the beginning. As the parcel of air is transported over the Atlantic Ocean and poleward, long chain PFCAs begin to form more quickly. The remaining intermediates at the end of this period have the potential to form much more PFOA and PFNA depending on the future fate of the air parcel. Despite emission into a very low-maximum yield environment, the transport is sufficiently fast to allow long-chain PFCA formation."   L326

[Figure]

Figure 5. Fluorotelomer precursor chemical fate along HYSPLIT trajectory through summertime photochemical environments. After two weeks, yields of PFOA and PFNA are approximately 1.5% and 0.7%, respectively, with more than 20% of the initial precursor still in an intermediate form which will undergo further reactions.

This approach is more complex than a box-model approach and poses difficulties of its own, but has advantages: Back-trajectories from select deposition regions (such as the Arctic) can be identified and traced back to source regions. The chemistry box model can then be operated with chemical and photochemical input from GEOS-Chem along the trajectories (thereby accounting for change in chemical composition along the trajectories). Thus, one can, in principle, calculate the overall yields on trajectories leading from select emission regions to select deposition regions. The transport issue would be mitigated (although mixing and non-chemical removal would still not be accounted for) and yield attribution to individual sources would become possible.

We do believe that a detailed version of this would be an interesting study to perform, but also

believe that it is outside the scope of this study. We have added an illustrative version of this concept, as described above.

The manuscript contains a critical flaw: A box model with fixed chemical conditions is used to investigate chemical processes that take place on time scales during which which chemical conditions are bound to change due to transport and mixing. I recommend a major revision only if the authors can compellingly demonstrate that the box model approach with fixed conditions is appropriate to investigate formation of PFCAs from from fluorotelomers, despite the formation taking place on time scales during which air parcels are transported and experience different chemical conditions.

We acknowledge that PFCA formation timescales are longer than the residence times in any given chemical environment, but would argue that it is informative to know the differences due to the environment themselves, particularly in how those differences compare to the uncertainty due to the reaction mechanism, as the reviewer points out above.

One way to demonstrate this would be to show that systematically using fixed chemical conditions gives, in reasonable approximation, the results one would obtain if realistic, changing conditions were used.

We do not believe that this would be the case, for the reasons that the reviewer points out in the above comments.

If this is not possible I recommend rejection in favor of a re-submission in which a more appropriate modeling approach, such as the outlined trajectory approach, is implemented.

Some detailed comments

For the benefit of the reader and to facilitate reproducibility, the below comments should be addressed and oversight errors corrected.

Section 2.1
- The numerical solver of the chemistry model should be briefly described.

We have added the sentence "We use the LSODE solver implemented in the scipy package of Python to solve the system of differential equations defined by this chemistry." L84

Section 2.2
- Diurnal cycle: Is it resolved in the simulations, or does the model use perpetual mean conditions, without diurnal cycle variation? Simulations resolving the diurnal cycle would be preferable, being more realistic, but if the latter approach was chosen: how were daily mean photochemistry rates calculated? Was the perpetual mean conditions approach tested by select simulations that do resolve the diurnal cycle, and what were the results? Such a test is inexpensive when a box or a trajectory model is

Used.

We used the perpetual conditions after testing with a diurnal cycle. The reason that mean
conditions provided such a similar response is that a) the chemistry involved scales with time of
day fairly uniformly, meaning that competing process tend to speed up and slow down together;
b) there are no non-chemical processes in our model competing with these processes that scale
together; c) there are no reactions in our mechanism or in the literature to our knowledge which
would be dominant during the nighttime.

- Actinic flux specification: A value of 1E15 photons cm-3 s-1 at 90 degrees solar zenith
angle is given (with reference to Seinfeld and Pandis, 2006). There are several issues
which should be addressed: The units of actinic flux are photons cm-2 s-1, not photons
cm-3 s-1. Solar zenith angle is measured from zenith: 90 degrees means the sun is
at the horizon. This is inconsistent with the specification "peak actinic flux", which,
in clear-sky conditions, occurs at noon (corresponding to a solar zenith angle that is
typically >= 0 but < 90).

Thank you for pointing out these two typos, we have changed 90 degrees to 0 degrees and
cm-3 to cm-2. L114

 Seinfeld and Pandis (2006, Table 4.3) give 340-365 nm mean
winter (5E14 cm-2 s-1) and summer (8.9E14 cm-2 s-1) noon actinic flux values at the
surface, at 40 degrees north. The actinic flux value used in the simulations is only
consistent with the summer value given in Seinfeld and Pandis (2006).

The peak values here are scaled by latitude and time of year to translate this peak flux for all
the photochemical environments.  We have changed the sentence to read "Available photons
for photolysis reactions were calculated based on a scaling by the position of the sun as a
function of latitude and time of year and an assumption of clear sky conditions (Russell), and
a peak actinic flux of 1x10^15 photons cm−2 s−1 at 0 degrees solar zenith angle (Seinfeld
and Pandis, 2006)." L114

- You write "Available photons for photolysis reactions were calculated as a function of
latitude and time of year ..." This should be explained in detail in the manuscript.

See previous comment.

Appendix A

- Units of the rate coefficients should be given.

We have added units to the rate coefficients table.

- Fluorotelomer aldehyde photodissociation: I tried to trace the rate coefficient for the
reaction 1 (Appendix A), for which the value 1.5+-0.75E-22 (no units) is given, with
reference to Young and Mabury (2010). Young and Mabury (2010) give two photodis-
sociation cross sections for FTALs, 13.3E-20 (no error estimate) cm2 (Chiappero et al.,
2006) and 5.4+-0.4E-20 cm2 (Solignac et al., 2007), at the maximum of the absorption
spectrum. Young and Mabury (2010) do not give the photodissociation rate coefficient.
How does the photodissociation rate coefficient 1.5+-0.75E-22 and its error estimate

arise?

The value of 1.5e-21 in the table is accounting for both the Solignac [2007] number that the reviewer quotes (5.4e-20) and the maximum full spectrum quantum yield of dissociation (0.04) from Sellevag [2004]. These are combined to relate the total actinic flux to the dissociation of FTAL molecules. The uncertainty estimate was increased to 50% because we apply the values to longer chain homologues of the actual molecules from these experiments, and because of the scarcity of data.

---

## Referee Report (RR1)

The approach taken in this paper has limitations that were made abundantly clear by the original reviews. However, the authors have done an acceptable job in replying to these criticisms, and have emphasized that aspects of the work that are new and have some utility. The paper should be accepted for publication.